# Comprehensive Knowledge and Preparedness among Dental Community to Confront COVID-19—A Multicentric Cross-Sectional Study

**DOI:** 10.3390/ijerph19010210

**Published:** 2021-12-25

**Authors:** Nada Faleh Almutairi, Amani Abdullah Almaymuni, Julie Toby Thomas, Toby Thomas, Abdullah Almalki, Roshan Uthappa

**Affiliations:** 1College of Dentistry, Majmaah University, Al-Majmaah 11952, Saudi Arabia; nada2.almutairi@gmail.com (N.F.A.); Ammman23i@gmail.com (A.A.A.); 2Department of Preventive Dental Sciences, College of Dentistry, Majmaah University, Al-Majmaah 11952, Saudi Arabia; ae.almalki@mu.edu.sa; 3Department of Restorative Dental Sciences, College of Dentistry, Majmaah University, Al-Majmaah 11952, Saudi Arabia; t.thomas@mu.edu.sa (T.T.); r.uthappa@mu.edu.sa (R.U.)

**Keywords:** COVID-19, dental community, infection control, awareness, preparedness, pandemic

## Abstract

The healthcare policy changes need to be updated for better management of the COVID-19 outbreak; hence, there is an urgent need to understand the knowledge and preparedness of healthcare workers regarding the infection control COVID-19. Therefore, the present study aims to assess the knowledge and preparedness towards COVID-19 among dentists, undergraduate, and postgraduates in dental universities one year after the COVID-19 outbreak. The multi-centric cross-sectional study was conducted by evaluating 395 structured, pre-coded, and validated questionnaires obtained from sample units comprising full-time dental students (undergraduates, interns, and postgraduates) and dentists who were currently in practice and who were able to comprehend the languages English or Arabic. The first part of the questionnaire included questions related to demographic characteristics. The second part of the survey consisted of questions that address knowledge concerning COVID-19. The third part of the survey addressed questions based on the preparedness to fight against COVID-19 including sharp injuries during this period. Comparing the knowledge scores of dentists, dental undergraduates, and postgraduates using the ANOVA test, dentists have higher knowledge and preparedness scores than postgraduates and undergraduates (*p*-value < 0.05). Univariate logistic regression analysis demonstrated that undergraduates and postgraduates were 2.567 and 1.352 times less aware of the personal protective measures against COVID-19 than dentists, respectively. Dentists had the comparatively better perception in knowledge and awareness of COVID-19 than undergraduates and postgraduates.

## 1. Introduction

Coronavirus disease (COVID-19) caused by severe acute respiratory syndrome coronavirus 2 (SARS-CoV-2) is a public health emergency of international concern. Direct mucous contact with saliva droplets, respiratory fluids, and aerosols are the main modes of viral transmission. Since the viral load contained in human saliva is very high, it may serve as a potential source of infection. Owing to the nature of the dental procedures and treatments, the dental office seems to be a high-risk environment for this nosocomial infection [1]. The breakout of the pandemic COVID-19, as well as the evolution of various mutant strains, has sparked research among global health professionals to limit the spread of the coronavirus by ensuring that preventive care norms and standards are regularly updated. 

To curb the spread of COVID-19 infection through dental clinics, the main focus should be diverted to investigate the level of knowledge and preparedness of the dental fraternity regarding the infection and how well they are prepared to fight against it. As a preparatory step, the World Health Organization (WHO) recommends that dental clinics institute routine pre-appointment triaging, temperature measurement, detailed health status investigation, and checking for COVID-19 risk factors, including recent travel and contact with an infected person [2]. Dental professionals are prone to potential exposure of the viral particles through direct face-to-face communications, exposure to saliva and blood, or indirect contact with the contaminated inanimate surfaces. Advocating various strategies to mitigate the transmission of infectious disease is of critical importance in the dental environment.

Assessment of preparedness regarding infection control against pandemics among the dental community is in line with the Saudi vision 2030 for fundamental structural changes in the healthcare sector to meet the growing demand for health care services in the kingdom [3]. Dentists, dental students, and dental assistants are among the frontline health workers in this pandemic period. Hence, to take measures against this pandemic, the dental fraternity should be aware and prepared to take several personal protection measures and minimize aerosol-generating operations as a step to prevent the spread of infection. 

Dental procedures can a pose high risk of viral transmission through the equipment used, which produces aerosols harboring high numbers of SARS-CoV-2 virions. Aerosol-Generating Procedures using dental turbines, micro-motor or rotary handpieces, ultrasonic scalers, and air-water syringes are speculated as maximum hazard categories for transmission of infection spread [1]. For this reason, routine dental care has been suspended throughout the COVID-19 pandemic, but the impact of long-span halt and restricting only to emergency treatment have affected the dental community psychologically and financially and the performance level in clinical practices [4]. 

Clinicians should be aware of aerosol mitigation strategies such as prior screening for identifying symptomatic patients, advocating pre-procedural mouth rinses such as 0.2% povodine iodine and 1% hydrogen peroxide, regular monitoring of dental unit waterlines (DUWLs), and use of high volume intraoral evacuation (HVE) before resuming full-fledged dental practice. Reports from an Italian study stated the effective use of telephonic triage to identify patients for urgent care. The increased optimistic attitude was observed with age towards the incorporation of digital technology and teledentistry among the practitioners, to identify emergency cases during pandemic situations [5]. 

Passarelli et al. in 2020 found that, during the emergence of COVID-19 infection, the preventive measures have escalated to higher levels according to the latest recommendations by the Centers for Disease Control and Prevention (CDC) and other global protection agencies [6]. In addition to the use of personal protective equipment (PPE) such as masks, respirators, and gowns and the use of air cleaning systems, surface decontamination, and airborne infection isolation rooms (AIIRs) are some structural modifications that can be advocated in dental units to minimize aerosol contamination [7]. The study by Rexhepi I et al. in 2021 demonstrated the effectiveness of natural ventilation and standard saliva ejectors by quantitatively evaluating the aerosol being generated during the dental procedures. They found that particulate material (PM) was increased during the dental procedures with open doors and windows. Further, low suction systems could reduce coarse (PM10) but were less effective for ultrafine particulate material (less than 1 micron in size), and the scaling procedure generated more particulate material compared to other dental procedures [8].

The SARS-CoV-2 infection has brought a new, unanticipated challenge to dental professionals. A recent study by Pylińska-Dąbrowska D et al. reported increased anxiety levels among patients undergoing oral surgical procedures during the pandemic period than before. This further highlights that dentists need to be prepared to face anxious patients in the future [9]. Research shows that, despite the positive attitude of dental professionals in Saudi Arabia and proper practice of droplet and airborne infections in similar outbreaks, there seems to be a lack of knowledge and attitude towards transmission-based precautions in general, which calls for proper training [10,11]. Al Jasser et al. in 2020 researched COVID-19 precautions among dental students at King Saud University, Riyadh, and found that they attained fair scores on knowledge and attitudes, but the common practice scores highlight the need for urgent strategies to prevent infection among the dental students, including mandatory crash courses and hands-on protection measures [12]. 

Despite several studies being conducted focusing on the pandemic preparedness regarding infection control against coronavirus disease among the dental community, there are still lacunae in the awareness of safety protocol. To facilitate healthcare policy changes for better management of the pandemic and its consequences in Saudi Arabia, there is an urgent need to understand the present scenario based on the knowledge and preparedness among the dental community regarding infection control against COVID-19 during these challenging times. 

In spite of regular updates from public health departments and the Ministry of Health regarding infection control, the literature still reports the existence of a void in knowledge and awareness among the dental community. This aroused our inquisitiveness to identify the areas for further improvement in the awareness and standards of infection control in the field of dentistry to confront COVID-19 among the dental professionals and students including undergraduates and postgraduates in the central region of Saudi Arabia. 

Therefore, this study aims to assess the knowledge and preparedness toward COVID-19 among dentists and dental students one year after the COVID-19 outbreak and to identify factors that need to be improved for pandemic preparedness. The study also focused to estimate the correlation between the knowledge of the dental community and their preparedness to confront the COVID-19 pandemic. 

## 2. Materials and Methods

A cross-sectional study was adopted for conducting a descriptive online structured survey during the period of 10th January to 30th March 2021 among the health workers related to the dental profession, belonging to any nationality from the central part of Saudi Arabia. The survey questionnaire in English or Arabic language was mailed to health centers, hospitals, and dental universities from randomly selected clusters in the region. The sample units comprised full-time dental students (undergraduates, interns, and postgraduates) and dentists who were currently in practice, having internet access, and those who were able to comprehend the languages English or Arabic. The subjects who did not consent to the study and worked on a part-time basis were excluded from the study.

The required sample size for this study was calculated based on Hajian-Tilaki (2011), where the significance level (alpha) was set to 0.05 and power (1-β) was set to 0.80. It resulted in a required final sample size of 384 individuals [13]. Therefore, 700 questionnaires were considered to minimize the errors and account for the drop-outs.

The ethical approval was obtained from the institutional ethics committee of Majmaah University, Saudi Arabia (Research Number: MUREC Dec.30/COM-2020/8-2; 18), in compliance with the Helsinki Declaration. Informed consent contained elaborate information about the study’s purpose and importance, subjecting the participants to autonomous decision making about whether to join or withdraw at any time. Subjects who opted to “agree” were instructed to complete the self-administered questionnaire. Confidentiality of the study subjects was ensured by avoiding collecting any personal details that included names, residence details, passport number, and history related to COVID-19 exposure. Only one response was permitted for an internet protocol (IP) address.

This web-based survey was formatted through Google Forms. The single-stage cluster sampling method was used for sample recruitment. The Google Forms link was circulated to the subjects through emails and WhatsApp groups among the student representatives, hospital/ health center administration department, and dental faculty affiliated with the universities. Out of 700 subjects to whom the questionnaire was sent, 159 disapproved of the submission of informed consent, and 289 forms were incomplete. The remaining 387 completed questionnaires were included for data analysis. The sample units were further classified into three groups—dentists, undergraduates, and postgraduates.

A standardized (structured, pre-coded, and validated) questionnaire was developed for this study by the investigators, and it is based on frequently asked questions (FAQ) found on the CDC and WHO official websites [10,14,15]. The questions were multiple choice and sought to gain insight into the respondent’s awareness and preparedness toward COVID-19. The content of the questionnaire was validated by four investigators to confirm the selection of scale items based on previously formatted questions. Each item was considered to be valid if more than two investigators approved it to be essential. After that, a pilot survey of 10 individuals from each group was undertaken after ethical approval was obtained to ensure that the questions elicited the appropriate responses and there are no problems with the entry of answers into the database. The validity of the questionnaire has been assessed and the reliability rating score was estimated to be 0.72 (Cronbach’s alpha).

The self-reported questionnaire comprised a total of 43 questions, which were subdivided into three sections. The first section consisted of 10 questions, designed to obtain information regarding demographic characteristics. The second section of the survey consisted of 19 questions that addressed the knowledge concerning COVID-19. The third section of the survey comprised of 14 questions that evaluated the preparedness to combat the pandemic period. The questionnaire was designed in English and was subsequently translated into Arabic after the ethical approval. It was pre-tested to ensure that it maintained its original meaning. 

The first part is designed to obtain background information, including demographic characteristics (nationality, age, gender, level of education, and occupation) and whether any training was received related to infection control and management of COVID-19 patients.

The assessment tool based on the knowledge concerning COVID-19 included questions based on the COVID-19 etiology: transmission mode, symptoms, diagnosis, and infection control. Each correct answer was assigned one point, and an incorrect/not sure answer had no points. The total knowledge score varied from 0 to 19, indicating the knowledge about the COVID-19 pandemic. 

Self-preparedness to tackle COVID-19 and dental sharp injuries was assessed based on the response to the tool that has been previously used and validated [11]. The questions were related to personal protection measures such as minimizing the use of public spaces, social gatherings, outdoor activities, and assessment based on good hygiene practices, such as hand washing, cleaning, sanitization, and vaccination as preventive measures. The behavior, if put into practice, was awarded one point, and 0 points were awarded if it was not practiced by the participant. The overall score ranged from 0 to 14, indicating the level of performance. 

Data representing the categorical variables were presented as frequencies and percentages. Comparison of each item in knowledge and preparedness scores was performed using the ANOVA test. Tukey’s post-hoc test was performed for inter-group comparison of knowledge and preparedness scores. Univariate logistic regression was computed using each item in knowledge and preparedness as an outcome separately to examine the relationships in the adjusted analysis. The relationship between knowledge and preparedness regarding infection control against COVID-19 was assessed using Pearson’s correlation analysis. A calculated *p*-value less than 0.05 was considered statistically significant. The questionnaire is available as Appendix A. All the analyses were carried out with the help of the commercially available statistical package SPSS v.23 for WINDOWS (IBM, Armonk, NY, USA).

## 3. Results

### 3.1. Demographic Characteristics of the Participants

Table 1 and Figure 1 and Figure 2 show the distribution of demographic characteristics of the study participants been expressed in frequency and percentiles.

### 3.2. Comparison of Knowledge Scores among the Participants

The mean knowledge scores of dentists, dental undergraduates, and postgraduates were found to be 15.76 ± 1.21, 10.54 ± 1.72, and 13.58 ± 1.47, respectively. Table 2, Figure 3 shows the comparison of knowledge scores of dentists, dental undergraduates, and postgraduates using the ANOVA test, and the results show that the difference was statistically significant with the *p*-value < 0.05 between three groups with dentists having higher knowledge and preparedness scores than postgraduates and undergraduates.

Tukey’s post hoc test was performed for the comparison of knowledge scores between groups. Table 3 shows that there is a statistically significant difference between each group. Three group comparison ANOVA has been used followed by a post hoc test for comparison between the inter-groups for knowledge.

### 3.3. Comparison of Preparedness Scores among the Participants

Table 4 and Figure 4 depict the mean preparedness scores of dentists, dental undergraduates, and postgraduates, which are 9.32 ± 1.87, 5.64 ± 1.45, and 7.68 ± 1.38, respectively. Comparing the preparedness scores of dentists, dental undergraduates, and postgraduates using the ANOVA test, we found that the difference was statistically significant with the *p*-value < 0.05.

A Tukey’s post hoc test was performed for the comparison of preparedness scores between groups. Table 5 shows that there is a statistically significant difference between each group, and dentists were found to be more prepared against COVID-19, followed by postgraduates and undergraduates.

### 3.4. Relationship between Participant’s Knowledge and Preparedness Scores

Pearson’s correlation coefficient was used to find the relationship between knowledge and preparedness regarding infection control against COVID-19. The results in Table 6 indicate the positive correlation between knowledge and preparedness, i.e., the preparedness scores increase with an increase in the knowledge scores.

### 3.5. Factors Affecting the Level of Preparedness against COVID-19

Univariate logistic regression analysis was performed to find the factors affecting the level of preparedness against COVID-19 among dentists, dental undergraduates, and postgraduates. Undergraduates and postgraduates were 2.567 and 1.352 times less aware of the personal protective measures against COVID-19 than dentists, respectively. This was found to be statistically significant. Dentists have twice more time preparedness and precautions to fight against COVID-19 than postgraduates and three times more time than undergraduates. Undergraduates were more poorly aware of the actions to be taken after needle stick injury than dentists and postgraduates. Table 7 shows that undergraduates are poorly aware of college sharp policy and procedures, i.e., two times less than the dentists.

## 4. Discussion

The level of ongoing community transmission of COVID-19 has paved the way to refine the thoughts of dental practitioners and the student community to resume their practice in a full-fledged manner. For the past year, the dental faculty, being aware that they are a high-risk category for spreading the virus through aerosol generating procedures from asymptomatic patients, have confined their practice to emergency or elective procedures.

Coulthard et al. in 2020 found that dentists play a vital role in preventing the transmission of COVID-19. They emphasized the need for organized urgent care delivered by teams provided with appropriate personal protective equipment. It was reported that dental professionals felt a moral duty to reduce routine maintenance for fear of spreading COVID-19 among their patients. However, the aftermath of COVID-19 has resulted in a profound negative impact on the socio-economic status and performance of dental clinical practices [16].

During the COVID-19 pandemic, the CDC had set infection prevention and control recommendations for dental operations [17]. Occupational Safety and Health Administration (OSHA) recommendations on Bloodborne Pathogens (29 CFR 1910.1030), Personal Protective Equipment (29 CFR 1910 Subpart I), and Respiratory Protection (29 CFR 1910.134) have been revisited and were sought to be applied in practice by the dental community [18]. It is mandatory that this standardized protocol is implemented regularly by all dental professionals before they resume their clinical practice.

Investigators have been focusing on assessing the level of knowledge and awareness among various population groups regarding the COVID-19 pandemic. National health undertakers around the globe have introduced COVID-19 guidelines and have taken vigorous steps in educating the health care providers by circulating health instruction manuals and guidelines on dealing with COVID-19 patients. Despite the efforts made, the literature still suggests existing lacunae in the knowledge of COVID-19 and preparedness amongst the dental community to confront the pandemic [19,20].

Since no previous study has been conducted focusing on the pandemic preparedness regarding infection control during COVID-19 among the dental community as a whole, a standardized (structured, pre-coded, and validated) questionnaire was developed for this study by the investigators, and it was based on frequently asked questions found on CDC and WHO official websites. The questions were multiple choice and sought to gain insight into the respondent’s awareness and preparedness towards COVID-19.

Therefore, this self-structured questionnaire study was undertaken to identify the area for improvement in knowledge and preparedness against the COVID-19 pandemic and to compare the factors affecting the preparedness among the respondents classified under dentists (44.7%), undergraduates (30%), and postgraduates (25.3%).

The demographic data of the study revealed that 9% of the study population still have not received any training related to infection control, and 34.1% of respondents are not trained to manage COVID-19 affected patients. The results of the study demonstrated that knowledge scores of the dentists were found to be higher than undergraduates and postgraduates and the difference between the scores was found to be statistically significant (*p* = 0.034). The findings are inconsistent with the study by Al Jaseer R et al. where the dental students from a university in Riyadh exhibited only a fair level of awareness about COVID-19 and knowledge of the safety measures during patient management, emphasizing the need to update on the current information and safety protocols on COVID-19 through online courses [12].

Inter-group comparison amongst the three groups based on the knowledge scores was also found to be statistically significant. The study by Rajeh M et al. in 2020 reported that the majority of dentists in the Mekkah region attained a good knowledge score regarding COVID-19. Moreover, nearly all of the participants demonstrated good attitudes towards precautionary measures in dental clinics. It was found that Saudi residents with a high education level are knowledgeable about COVID-19, hold favorable attitudes, and have appropriate practices towards precautionary measures needed while visiting a dental clinic during a virus outbreak [21].

Preparedness scores related to infection control were comparatively higher in dentists than undergraduates and postgraduates (*p* = 0.042). Tukey’s post hoc test to assess the inter-group comparison revealed statistically significant results, inferring that the dental students at the undergraduate level were least prepared to fight against the COVID-19 situation. In this study, Pearson correlation analysis demonstrated a significant positive association between preparedness and knowledge scores. Increasing knowledge among dentists would enable them to be more prepared to attend to COVID-19-infected patients seeking dental care. A Chinese online cross-sectional survey among the residents by Zhong et al. (2020) examined the knowledge, attitudes, and practices regarding COVID-19, and they concluded that the COVID-19 knowledge score was significantly associated with a lower likelihood of negative attitudes and preventive practices towards COVID-19. The majority of residents of good socioeconomic status who had adequate knowledge about COVID-19 demonstrated optimistic attitudes and followed appropriate practices towards the pandemic [22].

Comparing the factors affecting the level of preparedness, it was found that undergraduates and postgraduates students were 2.57 and 1.35 times less aware than dentists of personal protection measures against COVID-19. Dentists were comparatively prepared to confront the pandemic situation twice more than postgraduates and thrice more than undergraduates. Studies have concluded that dental practitioners in the kingdom are well aware and prepared with the latest COVID-19 updates and preventative measures through Ministry of Health (MOH) guidelines and recommendations. However, several dentists reported being unaware of clinical–patient management methods. This can be resolved by organizing educational sessions, which need to be constantly monitored to ensure strict adherence to the updated MOH guidelines. A contradictory study by Batra K et al. in 2021 demonstrated that the social behavior of the dental students in India correlated well with their level of knowledge on the pandemic [23].

Reported evidence states that dental professionals are more vulnerable to sharp injuries, where the most common reason was found to be anesthetic needles, followed by hand scalers, suture needles, orthodontic wires, and endodontic files [24]. Poor awareness existed among the undergraduates of the college’s sharp policy and procedures and about the management of needle stick injury. The results of the study contradict the findings by Alharabi TM et al. in 2021, where they stated that, in spite of having good knowledge and a positive attitude towards standard infection control protocol, they still lack sufficient knowledge concerning preventive measures to combat disease transmission in dental clinics [25].

In light of the current evidence, it can be concluded that undergraduate students demonstrated insufficient awareness and preparedness compared to dental professionals. This could be probably due to the lack of inadequate patient interaction and accessibility to continuing education programs on infection control and safety measures. Therefore, the study emphasizes the need to focus upon training the dental student category on the importance of infection control, taking preventive steps to confront COVID-19 infection, and introducing awareness to college sharp policy and procedures.

The survey conducted by Arora et al. (2020) in India assessed the knowledge, risk perception, attitude, and preparedness of the dentists about COVID-19 among 765 dentists and concluded that most dentists had a fair knowledge about the characteristics of COVID-19 [26]. In a study, Al-Khalifa et al. (2020) found that there were varied dentist perceptions of the COVID-19 pandemic depending on demographic variables such as age and years of work experience [3]. The limitation of our study is that the difference in perception and preparedness level was not analyzed among dentists based on experience in the field of dentistry. Hence, further longitudinal studies are warranted to determine their preparedness based on their clinical experience.

## 5. Conclusions

Dentists had the comparatively better perception in knowledge and awareness of COVID-19 than undergraduates and postgraduates. The questionnaire included an overall perspective on the mode of spread of COVID-19, potential sources of viral contaminants in the field of dentistry, its symptoms, and different methods to stay protected. The study also assessed their knowledge of strategies to control aerosols in the dental environment as well as their preparedness in dealing with needle injuries. The results of the study further instigate the necessity to reform educational curricula for preparing the students to deal with COVID-19 and pandemic conditions in the future. The risk of reducing nosocomial infection can be enhanced by introducing training programs on infection control among dental students in the region. However, certain limitations that could be pointed out in this study include survey questions related to remote triage diagnosis by implementing teledentistry to every practice, and assessing knowledge related to monitoring and maintenance of DUWL could be included. Further studies from this region are warranted on these aspects.

## Figures and Tables

**Figure 1 ijerph-19-00210-f001:**
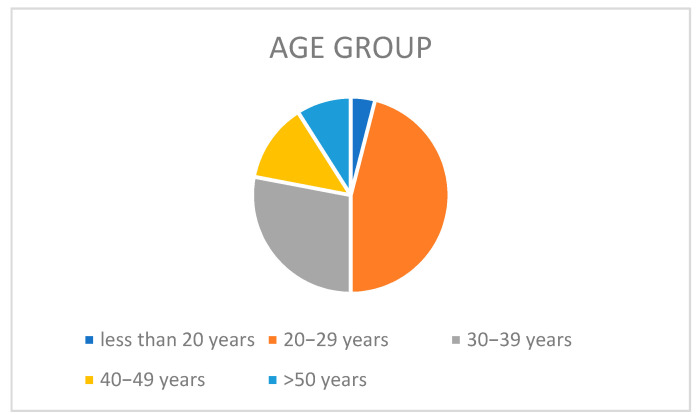
Distribution of respondents according to age.

**Figure 2 ijerph-19-00210-f002:**
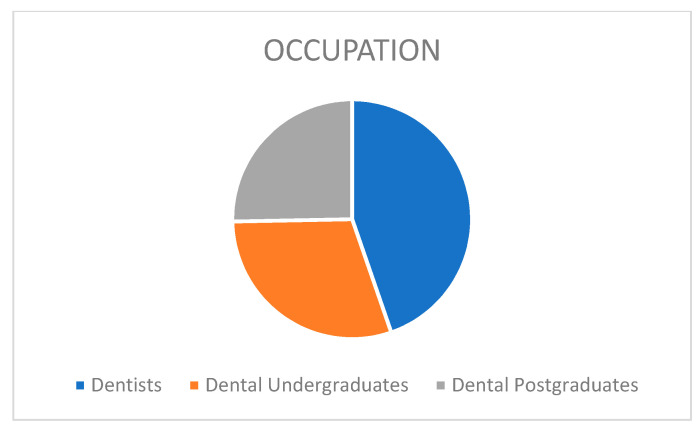
Distribution of respondents belonging to dental community.

**Figure 3 ijerph-19-00210-f003:**
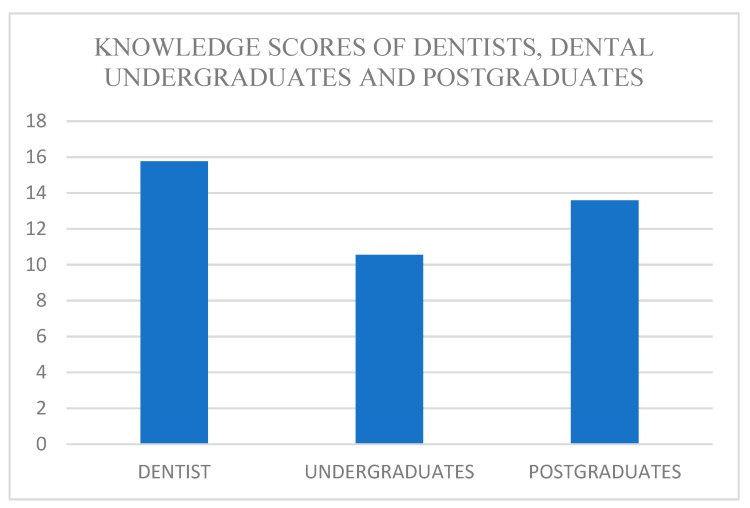
Knowledge scores of dentists, dental undergraduates, and postgraduates.

**Figure 4 ijerph-19-00210-f004:**
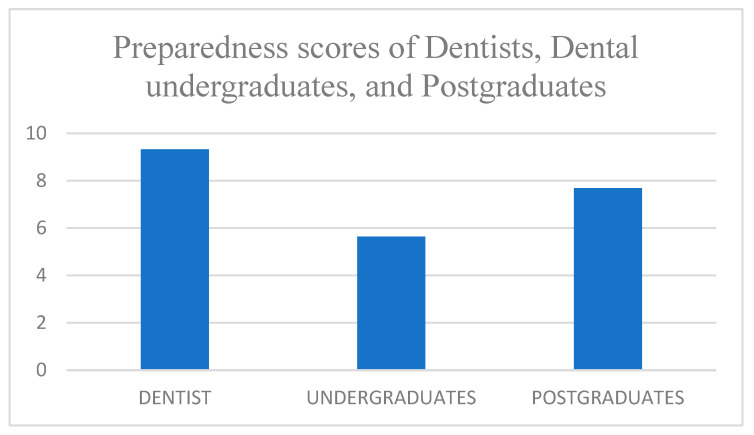
Preparedness scores of dentists, dental undergraduates, and postgraduates.

**Table 1 ijerph-19-00210-t001:** Distribution of participants based on demographic characteristics.

Characteristics	Distribution of Respondents Frequency (*n*) 387 Percentage (%)
Age group	
<20 years	16 (4.1)
20–29 years	177 (45.7)
30–39 years	110 (28.4)
40–49 years	50 (12.9)
>50 years	34 (8.8)
Gender	
male	194 (50.1)
female	193 (49.9)
Duration of practice	
<5 years	172 (44.4)
5–15 years	154 (39.8)
>15 years	61 (15.8)
Area of practice	
Urban	374 (96.6)
Rural	13 (3.4)
Occupation	
Dentists	173(44.7)
Dental undergraduates	116(30)
Dental postgraduates	98(25.3)
Training taken related to infection control	
Yes	352 (91)
No	35 (9)
Training taken related to manage COVID-19 patients	
Yes	255 (65.9)
No	132 (34.1)

**Table 2 ijerph-19-00210-t002:** Knowledge Scores of dentists, dental undergraduates, and postgraduates.

Mean Knowledge Score				
Dentists	Dental Undergraduates	Dental Postgraduates	F Value	*p* Value
15.76 ± 1.21	10.54 ± 1.72	13.58 ± 1.47	3.36	0.0343 *

(ANOVA test) (* denotes statistically significant result).

**Table 3 ijerph-19-00210-t003:** Intergroup comparison of knowledge score between dentists, dental undergraduates, and postgraduates.

Mean Knowledge Score	Dentists	15.76 ± 1.21	0.0247 *
	Dental undergraduates	10.54 ± 1.72	
	Dental undergraduates	10.54 ± 1.72	0.0321 *
	Dental postgraduates	13.58 ± 1.47	
	Dentists	15.76 ± 1.21	0.0472 *
	Dental postgraduates	13.58 ± 1.47	

(Tukey’s post hoc Test) (* denotes statistically significant result).

**Table 4 ijerph-19-00210-t004:** Preparedness scores of dentists, dental undergraduates, and postgraduates.

Mean Preparedness Score				
Dentists	Dental Undergraduates	Dental Postgraduates	F Value	*p* Value
9.32 ± 1.87	5.64 ± 1.45	7.68 ± 1.38	3.08	0.0421 *

(ANOVA test) (* denotes statistically significant result).

**Table 5 ijerph-19-00210-t005:** Intergroup comparison of preparedness score between dentists, dental undergraduates, and postgraduates.

Preparedness	Dentists	9.32 ± 1.87	0.0271 *
	Dental undergraduates	5.64 ± 1.45	
	Dental undergraduates	5.64 ± 1.45	0.0356 *
	Dental postgraduates	7.68 ± 1.38	
	Dentists	9.32 ± 1.87	0.0331 *
	Dental postgraduates	7.68 ± 1.38	

(Tukey’s post hoc Test) (* denotes statistically significant result).

**Table 6 ijerph-19-00210-t006:** Relationship between knowledge and their preparedness regarding infection control against COVID-19.

	Knowledge r^2^	Preparedness r^2^
Knowledge	1.00 (0.00)	0.687(0.024 *)
Preparedness	0.687 (0.024 *)	1.00(0.00)

(Pearson’s correlation coefficient) (* denotes statistically significant result).

**Table 7 ijerph-19-00210-t007:** Comparison of factors affecting level of preparedness among dentists, undergraduates, and postgraduates.

Factors	Occupation	Odds Ratio	*p*-Value	Confidence Interval	
				Upper Limit	Lower Limit
Personal protective measures	Dentists	Ref			
	Postgraduates	1.352	0.046 *	1.008	1.686
	Undergraduates	2.567	0.031 *	1.958	3.023
Preparedness to fight against COVID-19	Dentists	Ref			
	Postgraduates	2.301	0.037 *	1.745	2.783
	Undergraduates	3.528	0.028 *	2.851	3.848
Infection control	Dentists	Ref			
	Postgraduates	1.511	0.034 *	1.021	2.101
	Undergraduates	2.65	0.026 *	2.158	3.174
Precautions against COVID-19	Dentists	Ref			
	Postgraduates	2.20	0.021 *	1.978	2.892
	Undergraduates	3.58	0.018 *	3.011	4.032
Actions taken after a needle stick injury	Dentists	Ref			
	Postgraduates	2.03	0.025 *	1.478	2.478
	Undergraduates	3.79	0.018 *	3.022	4.134
College sharp policy and procedures	Dentists	Ref			
	Postgraduates	1.352	0.038 *	0.891	1.731
	Undergraduates	2.86	0.023 *	2.412	3.453

Univariate logistic regression (* denotes statistically significant result).

## Data Availability

Data supporting reported results can be presented on request.

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
