# Peer review of "Comprehensive Knowledge and Preparedness among Dental Community to Confront COVID-19—A Multicentric Cross-Sectional Study"

_ijerph, 2021, doi:10.3390/ijerph19010210_

Round 1
Reviewer 1 Report
The article entitled “Comprehensive Knowledge and Preparedness Among Dental Community to Confront COVID 19 A Multicentric Cross-sectional study” aimed to investigate the knowledge and preparedness toward COVID-19 among dentists, undergraduate and post-graduates in dental universities one year after the COVID-19 outbreak. Authors have well revised several issues; however, I ask authors to add some key concepts.
- Authors must discuss more on COVID-19 pandemic, specifically on some dental-related aspects (Please, see and discuss doi:10.3390/ijerph17165780). In fact, other surveys were conducted aimed at dental practitioners to evaluate the behavior of dentists during the COVID19 lockdown period and the feeling of fear regarding their professional future. Moreover, it would be interesting evaluating other findings regarding PM particles released during dental procedures due to the high risk of SARS-CoV-2 transmission of dental-care workers (please, see and discuss DOI: 10.3390/ijerph18147472) in order to reduce the production of PM particles produced by the use of airborne spreading devices that can act as vectors of infectious agents and contaminate dental practitioners, patients, and working surfaces in the operating space.
- The part of materials and methods is well described
Minor issues:
- Conclusions cannot be reduced to a sentence: you must improve them highlighting the limits and the future insights pointed out from this article.
- Several moderate typos are present in the text, please, amend
- The references must be reformatted according to the instructions for the authors (see Journal Articles: Author 1, A.B.; Author 2, C.D. Title of the article. Abbreviated Journal Name,Year, Volume, page range.)
Author Response
Sir,
Thank you for your valuable comments. I have made the necessary changes in the manuscript as per your comments. Kindly see the attachment.
Thanking you
Julie Toby Thomas

Reviewer 2 Report
Dear Authors,
The article: 'Comprehensive Knowledge and Preparedness Among Dental Community to Confront COVID 19 A Multicentric Cross-sectional study' was to assess the knowledge and preparedness toward COVID-19 among dentists, dental students one year after the COVID-19 outbreak, and to identify factors that need to be improved for pandemic preparedness. The study also focused to estimate the correlation between knowledge of the dental community and their preparedness to confront COVID -19 pandemic.
English language and style must be corrected.
Punctuation mistakes should be corrected.
Add this article in introduction: https://doi.org/10.3390/jcm9103344
p value is written in italics.
Add a table with abbreviations.
References should be prepared in accordance with the MDPI guidelines
Add the research form in the additional materials.
To sum up, article should be reconsider after minor revision.
Author Response
sir,
Thank you for your valuable suggestions. I have made the necessary changes in the manuscript as well as attached the table of abbreviations and research form as an additional file.
Thanking you
Regards
Julie Toby Thomas
